# Physical Activity Behavior, Motivation and Active Commuting: Relationships with the Use of Green Spaces in Italy

**DOI:** 10.3390/ijerph19159248

**Published:** 2022-07-28

**Authors:** Alessia Grigoletto, Alberto Loi, Pasqualino Maietta Latessa, Sofia Marini, Natascia Rinaldo, Emanuela Gualdi-Russo, Luciana Zaccagni, Stefania Toselli

**Affiliations:** 1Department of Biomedical and Neuromotor Sciences, University of Bologna, Via Selmi 3, 40126 Bologna, Italy; alessia.grigoletto2@unibo.it (A.G.); stefania.toselli@unibo.it (S.T.); 2School of Pharmacy, Biotechnology, and Sport Science, University of Bologna, 40126 Bologna, Italy; alberto.loi10@studio.unibo.it; 3Department for Life Quality Studies, University of Bologna, 47921 Rimini, Italy; pasqualino.maietta@unibo.it (P.M.L.); sofia.marini2@unibo.it (S.M.); 4Department of Neuroscience and Rehabilitation, Faculty of Medicine, Pharmacy and Prevention, University of Ferrara, Corso Ercole I d’Este 32, 44121 Ferrara, Italy; rnlnsc@unife.it (N.R.); emanuela.gualdi@unife.it (E.G.-R.)

**Keywords:** active commuting, green urban space, motivation, physical activity, COVID-19

## Abstract

Many benefits of physical activity (PA) are observed with weekly average volumes of 150–300 min at moderate intensity. Public parks may be an attraction for many people living in the city and could help to achieve the recommended dose of PA. The present study aims to understand the motivation that drives people to a park and evaluate the amount of PA practiced by park-goers. A questionnaire was anonymously administered to 383 voluntary visitors to the Arcoveggio park (Bologna), aged 18–70 years. Sixty-one percent of participants practiced outdoor PA. Differences in park use between sexes and age groups were found. PA was higher in men than in women and in the 18–30 age group than in other age groups. Most participants travelled to the park in an active way (86.4%), resulting in easier attainment of the recommended amount of PA (64.5%). The main motivations for using the park were related to relaxation, performing PA, or both. According to a multiple regression model, the time per week spent at the park, the method of getting there, and the kind of PA were significant explanatory variables of the amount of PA practiced. In particular, the highest number of minutes of PA was achieved by those who travelled to the park by running, while those using vehicles presented the lowest number. All initiatives to promote active commuting and activities in the urban park represent an important strategy to improve health, supporting adults to lead an active lifestyle.

## 1. Introduction

The World Health Organization guidelines on physical activity (PA) to avoid sedentary behavior [1] support the finding that many of the health benefits of PA result from average weekly volumes of 150–300 min of moderate intensity or 75–100 min of vigorous intensity, or an equivalent combination of the two. Promoting PA is a well-established priority, since physical inactivity is one of the five leading global risks for mortality in the world [2]. Indeed, physical inactivity is believed to be responsible for the death of 3.3 million people annually worldwide [3,4]. Furthermore, the recent emergence of SARS-CoV-2 has influenced the lifestyle of the population, reducing PA and becoming a serious concern, mainly for older adults who are typically more prone to chronic diseases and less active, compared to younger people [5,6,7]. Strategies are therefore needed to increase PA and reduce the sedentary lifestyle of the population. One of the strategies that could help to achieve the goal of the recommended levels of PA could be active transport such as walking or bicycling from home to work, shopping, recreational places, and vice versa [8]. In this light, a suitable plan for promoting PA [9], maintaining a healthy weight [10], and improving mental health [11,12] may be to empower citizens to switch from using private motor vehicles to active transportation. In addition, public transportation options (e.g., buses or trains) can encourage people to walk from and to various public transportation stops, increasing their PA levels, albeit to a lesser extent. These good habits could consequently also lead to benefits for the urban environment, decreasing pollution, traffic noise, and temperature [13]. A worldwide study showed that PA levels are higher in walkable cities [14], because they allow active commuting and allow more frequent to travel from home to downtown or other destinations within the city by bicycle or on foot. According to Zijlema et al. [13], the increase of PA may be most successful when integrated into daily life habits. Another factor that can help people to achieve the right amount of PA is the use of green parks. Public green spaces provide multiple health benefits by facilitating PA, contact with nature, and social interaction [15,16,17,18,19,20]. In addition, outdoor exercise can be a viable alternative to indoor exercise; exposure to a natural environment is linked to triggering a higher amount of PA among residents, and a lower mortality rate [21,22,23]. Some studies have showed that long-term adherence to exercise initiatives conducted in an outdoor natural environment or urban green space may be superior to that of indoor exercise interventions [16,24,25]. Despite several pieces of evidence on the health benefits of parks, they are generally underutilized, and visitors are often engaged in low levels of PA during their park visits [19,26]. Initiatives created to increase PA in green spaces have been linked with improvements in social networking and feelings of connectivity and companionship, an increased appreciation of nature, improvement in self-esteem, and a means of escape from modern life [27,28]. Even though there is increasing literature about the practice of PA, and on the motivations and interventions related to the increase in outdoor PA in green spaces, few studies have been carried out in Italy [29]. In addition [30], the motivations that cause people to use green space are still unclear [31,32] and, therefore, it is important to understand how these can be linked to PA practice. As regards the use of parks, some concerns are about the best distance from residences to urban parks and greenness to ensure a frequent use of green space. The currently recommended residence distance to the nearest city parks is 300 m [33]; however, other studies have suggested that people are willing to walk for an even longer distance to have access to a green urban space if parks have some attractive features [34,35]. Understanding the motivations that drive people to use green urban space is important to implement adequate strategies.

The purposes of this study were: (1) to assess people’s motivations to use the park, (2) to assess how many people in each sex and age group use the park to do PA, and therefore to understand how much park use affects PA levels, and (3) to evaluate the contribution of active, vehicle-free transportation (walking, jogging, bicycling) in achieving of the recommended levels of PA. To achieve these goals, a new questionnaire designed ad hoc was developed and administered in Arcoveggio park, in Bologna (North Italy), which is the capital of the Emilia-Romagna region, with nearly 4.4 million inhabitants over an area of 22,446 km^2^ [36]. Arcoveggio park covers nine hectares in the city‘s northern sector. The park contains outdoor fitness equipment, picnic areas, trails, and bicycle paths. As the COVID-19 pandemic has changed the lifestyle habits, this research could be a starting point to understanding the motivation that leads people to use the park and plan interventions to increase PA.

## 2. Materials and Methods

### 2.1. Questionnaire Development and Procedures

A new questionnaire was developed. To design the questionnaire, we drew on our personal experience and previous studies in the literature [13]. The items were independently submitted to the opinion of three researchers with expertise in PA to assess their clarity and relevance. The questionnaire, administered anonymously, and was divided into two sections (see Appendix A): the first section was designed to collect demographic information, including age, sex, weight, height, profession, and level of education. The Body Mass Index (BMI, kg/m^2^) was calculated from the referred values of weight and stature, and the weight status was assessed according to the World Health Organization guidelines [37]. The second part of the questionnaire consisted of 15 questions designed to assess important motivation to use the park, and the quality and amount of PA practiced. The questionnaire was written and administered in the Italian language; however, in Appendix A it is reported in English. The total amount of PA and the time of active commuting were calculated by multiplying the active time and the journey time by the number times a week that the participants visited the park. Finally, the two amounts were summed to calculate the total amount of PA and of active commuting.

### 2.2. Participants

Three hundred eighty-three individuals were randomly recruited among park-goers. The same researcher administered a printed questionnaire to all participants at Arcoveggio Park in Bologna (North Italy). The park, which is one of the largest parks in the city, is located in a neighborhood of socio-economic variability, representing, therefore a cross-section of the population of Bologna [38]. Questionnaire administration began in March 2021, during the pandemic, and was completed in April 2021.

The study included a sample of men and women who met the following inclusion criteria: having signed the informed consent; being a park-goer; and aged between 18 and 70 years. Pregnant women were excluded from the study.

The survey was approved by the Bioethics Committee of the University of Bologna (prot. N. 0224254 of 9 October 2020).

### 2.3. Statistical Analysis

The internal consistency of the questionnaire was evaluated by Cronbach’s alpha coefficient on the answers of the recruited sample. Cronbach’s alpha is considered reliable for values between 0.5 and 0.9. In addition, a test–retest method was used to assess the reliability of the questionnaire 15 days later.

Subsequently, to better achieve the objectives of the study, we performed an a priori power analysis using G*Power (version 3.1.9.2, Universität Kiel, Kiel, Germany) to determine sample size, given α, power, and effect size. When ANOVA was performed (α = 0.05; 1-β = 0.95; effect size f = 0.25), a sample size of 303 participants was detected. The outcomes parameters for the multiple regression detected a sample size of 123 participants. Additional subjects were involved to ensure the availability of data in the case of problems with data collection. Variables’ normality was verified with the Shapiro–Wilk test. Descriptive statistics (mean and SD for continuous traits, and frequency for discontinuous traits) were calculated. Differences in frequency distribution between groups were evaluated by the Chi-squared test. Two-way ANOVAs were carried out to assess differences among sexes and age classes in anthropometric characteristics and questionnaire items. When a significant F ratio was obtained, the Tukey post hoc test was used to evaluate the differences between the groups.

Finally, a multiple regression analysis was carried out to assess possible predictors of the amount of PA. Before performing the multiple regression, all the assumptions were verified. The Shapiro–Wilk test and the variance inflation factor (VIF) test were performed to verify the normal distribution and the multicollinearity of the variables. Anthropometric and sociodemographic variables and information regarding the use of the park were included in the model as independent variables. Predictors inputted into the model were those found to have significant associations with the total minutes of PA (i.e., *p* < 0.05). The data analysis was performed using Statistica for Windows, version 8.0 (Stat Soft Italia Srl, Vigonza, Padua, Italy).

## 3. Results

The questionnaire was validated using the test–retest method and Cronbach’s alpha was used to provide a measure of the internal consistency. The value was 0.70 which is acceptable. Twenty-five people were asked to complete the questionnaire twice, at a distance of two weeks, in order to assess the reliability of the survey. The correlation values are presented in Table 1.

Most of the respondents were females (*n* = 215, 56.1%). Since the age range of the participants was wide (from 18 to 70 years), people were divided into 10-year age class groups, with the exception of the first group (first group: 18–30 years). The class most represented was the first (*n* = 130, 34%), followed by those aged 31–40 years (*n* = 75, 20.1%), 41–50 years (*n* = 67, 19.6%), 51–60 years (*n* = 65, 17%), and 61–70 years (*n* = 46, 12%).

Table 2 summarizes the anthropometric characteristics of the study participants.

As expected, men had significantly higher mean values of weight and height (76.1 ± 11.8 kg and 176.9 ± 7.3 cm) than women (63.9 ± 13.1 kg and 164.3 ± 7.6 cm) (*p* < 0.001). In both sexes, the youngest age class had the lowest mean weight values, while males of the oldest age class and females of the age class of 51–60 years had the highest values. Regarding BMI, both women and men presented a significantly higher incidence of overweight and obesity with age (*p* < 0.001 for both women and men). In particular, the men belonging to the age class 61–70 years had the highest value while those belonging to the age class of 18–30 years had the lowest. Females showed higher frequencies of underweight than males, but also of obesity. An exception is represented by the oldest age group, where males showed a higher prevalence of overweight and obesity.

Table 3 summarizes the demographic and socio-economic characteristics of the subjects who participated in the study and the categorical questionnaire items. Most of the participants were employed (n = 285, 74.4%), followed by students (n = 45, 11.7%), retired (n = 37, 9.7%), and, finally, unemployed (n = 16, 4.2%). A large proportion of the sample reached the park by walking (n = 151, 39.4%), followed by running (n = 86, 22.5%), using a motor vehicle (n = 69, 18%) or a bicycle (n = 43, 11.1%). Most of the participants preferred outdoor PA (88.9%), perhaps because outdoor PA was considered safer than gym training during the COVID-19 pandemic [39,40].

Significant differences among the age groups separately by sex were observed in all items, except for Q12, “If you practice indoor exercise, in which type of indoor environment?” in both sexes, for Q5, “Do you go to the park to practice PA?” in females and, for Q6, “If you do not practice PA, why do you go to the park?” in males. Regarding the way to reach the park, females of all ages preferred to walk and secondarily to use a motor vehicle; males also preferred to walk and using a vehicle was their last preference. Men of all age classes frequented the park to practice outdoor PA, except the oldest ones. Regarding the kinds of exercise practiced in the park, women showed an increase in walking with an increase in age, while men presented a bigger variability among the different age classes. With increasing age, both sexes preferred to practice outdoor PA instead of indoor PA. To the Q6, “why do you go to the park?” young women generally answered that they go to the park to relax or to socialize, but increasing age increased the percentage of women who go to the park to get in touch with nature or to relax, while the percentage of those who go there to socialize decreased. For men, the percentage of those visiting the park to relax decreased with the increasing age.

In particular, considering the sub-categories, some differences emerged. As regards occupation, differences between sexes were observed for all the categories. In particular, in females there were fewer employed subjects in the youngest and oldest age categories and a greater number of students in the youngest. No significant difference was observed between sexes in educational level, and, as expected, the distribution of different levels of education differed significantly among each age class. For Q1, the differences between sexes were always significant, except for running; significant differences were almost always observed among age classes, but the majority of people preferred walking. The use of the park to do physical activity (Q5) differed between sexes, as the majority of males went to the park to do PA at every age, as opposed to females. No differences were observed between sexes for Q6 because the majority went to the park to relax; differences among age classes were more marked in females. The kind of PA practiced (Q7) differed between sexes because the majority of females preferred walking, whilst men gave more heterogeneous results. Q10 did not show differences between sexes, while differences were observed among age classes, as with increasing age, more subjects would not have practiced PA indoors. No significant differences were shown for Q12, both between sexes and age classes.

In addition, since the distance between home and the park could be an important factor that influences the decision on the way to reach the park, a Chi-squared test was conducted between these two variables. The distance was divided into the following five categories: less than 300 m, from 300 m to 1000 m. from 1000 m to 2000 m, from 2000 m to 4000 m and over 4000 m. The *p*-value of the Chi-squared test was statistically significant for the total sample (0.00) and for all the other subcategories.

The results of the two-way ANOVAs to evaluate sex and age group differences are reported in Table 4.

Significant differences were found in Q10, “How often would you practice indoor PA if there wasn’t a park (h/week)?”, in the preference to practice indoor PA, the level of satisfaction with the park, and in the level of fatigue when reaching the park. From what the question, “How often would you practice indoor PA if there wasn’t a park (h/week)”, the respondents of age class 51–60 years would have practiced little activity, while the youngest ones would have practiced it anyway. Participants in the age class 31–40 years showed the highest preference for indoor PA. Most participants in all the age groups were satisfied with the park, but the most satisfied were those in the age class 41–50 years. The respondents in the age class 18–30 years took the longest time to reach the park, because they came from more distant places. Regarding sexes, significant differences were found in levels of fatigue when reaching the park: women reported higher levels of fatigue than men. The oldest men used the park more often (3.75 ± 1.94 times a week), while the oldest women used it less (2.33 ± 1.22 times a week). Generally, the participants felt more energetic and more peaceful after visiting the park.

Of the total of 383 participants, 232 (60.6%) usually practiced outdoor PA. Figure 1 shows the amount of PA practiced in the park and active commuting to reach the park by sex and age classes. The figure shows that, generally, men practiced more PA than women, except for the age group 61–70 years, in which women practiced more PA than men. However, only men in the age class 18–30 years achieved the goal of 150 min/week of moderate PA, on average, while men in the age class 61–70 years showed the lowest PA level.

Considering only the amount of PA in the park, only seven participants (3.0%) achieved the goal of the 150 min/week moderate PA. On the other hand, if the time to actively reach the park, through walking, running, or bicycling, was considered as a part of PA, the amount of PA increased, and the participants who achieved the goal of 150 min/week increased to 118 (64.5%). Most of those who achieved the goal were men (n = 70, 59.3%), and the most represented age class was 18–30 years (n = 50, 42.4%). Concerning people who did not achieve 150 min/week, the largest number of these people reached the park by walking, running or bicycling (n = 95, 83.3%), and only a few people used motor vehicles (n = 19, 16.7%). Fifty-one percent of these participants were women (n = 59) and were in the age class 18–30 years (n = 31, 27.2%).

A multiple regression model was carried out to quantify the relationship between the dependent variable (total minutes of PA including active commuting) and the explanatory variables. The VIF was less than 10 for all the variables considered, so there was no multicollinearity. The results of the multiple regression are shown in Table 5.

This model explained 14% of the variance. The number of visits per week tot the park, reaching the park by running, and skating or walking at the park showed a positive relationship with the total minutes of PA, while using a vehicle (car, scooter, or public transport) showed a negative relationship.

## 4. Discussion

The purposes of the present study were to assess people’s motivations to use the park, and among the motivation, particular attention should be paid to PA, to understand how much park use affects PA levels. The final purpose was to evaluate how active vehicle-free transportation (walking, jogging, bicycling) influences the achievement of the recommended levels of PA. These aspects have become particularly important in relation to the lifestyle changes imposed by COVID. Regarding the participants of the present study, there were slightly more women than men. This is in accordance with the results of other studies [41] which have reported that women have a greater willingness to participate in surveys than men and have a greater engagement with the neighborhood environment [42,43,44]. Regarding age classes, the oldest people presented the highest values in weight, BMI, and overweight/obesity, highlighting the greatest health risk of these groups, since overweight/obesity is a potential risk factor for the occurrence of cardiovascular diseases [2], although their involvement in PA is a healthy habit to be maintained and strengthened. The youngest age class is the group that took the most time to reach the park. This is in accordance with previous studies, which have suggested that young people are willing to walk for longer distances than the recommended 300 m, to have access to green urban space, if parks have some attractive features [34,35]. At the same time, the age class 18–30 years was the most likely to carry out PA, regardless of where it takes place. Regarding the interaction between age classes and sexes, it is noteworthy that men in the age class 61–70 years were the ones who used the park the most. According to previous studies, people of the oldest age group usually have a better perception of the green urban space and often spend their leisure time in this kind of environment [45]. On the contrary, in the present study, women aged 61–70 years had the lowest score for time spent in the park. These data are in contrast with the study by de Vries et al. (2003), which found that women over 65 generally showed a higher frequency of use of a park, in comparison with men or people belonging to other age groups [46]. This result could be linked to the COVID-19 pandemic situation, since the oldest people could have been more afraid about going out and visiting public spaces, due fear of becoming infected with the virus.

As regards the study’s first purpose, the two main motivations to reach the park were to relax and to practice PA. The opportunity to perform PA, by promoting leisure walking, walking through the space when running errands, active playing, and sports, is another mechanism that has been proposed to explain the beneficial effects of a green environment [4,15,44,45,46]. Several studies have observed the efficacy of outdoor PA, but it is still unclear what might be the best kind of PA [47]. For this reason, we investigated the relationship between BMI and PA, since the results are not consistent in the literature. While some studies have suggested an inverse relationship between BMI and PA [48,49,50], other studies have demonstrated a weaker association [51,52]. The present study did not show any relationship between the two parameters (*p* value = 0.10), suggesting that the practice of PA is independent of BMI.

Another motivation to reach the park was to relax. Although there is increasing literature about the beneficial effects of the outdoor natural environment, the mechanisms that explain this relationship are still unclear. Thinking of the park as a place in which it is possible to relax is consistent with the “restoration theory”, which explains the beneficial effects by the intrinsic quality of the natural outdoor environment. So, health perception and well-being are influenced by watching a green space [3,49,53,54]. The results of the present study could be linked to other studies that have found that short-term exposure to forests, urban parks, gardens, and other natural environment reduces stress and depressive symptoms, restores attention fatigue, increases self-reported positive emotions, and improves self-esteem, mood and perceived mental and physical health [34,53,54,55,56,57,58,59]. The result relating to the use of the park to relax is important because it highlights how the population perceives the use and the benefits of this park. It can be considered a “safe place” in which they can stay and relax without other problems or thoughts, and it demonstrates the success of the project carried out by the city of Bologna to improve the use of green urban space [60]. After visiting the park, the largest part of the sample reported positive sensations, such as feeling “more energetic”, and “more peaceful”, which are in accordance with previous studies that observed a beneficial association between exposure to green space and mental health, using a wide range of measures [35,60,61,62,63,64,65,66]. The largest proportion of the participants who used the park to relax were women and people belonging to the youngest age class group (18–30 years). The effects of urban parks could be especially important for women [65] because they are disproportionately affected by common mental health issues [40].

Regarding the second purpose of this study, 232 (60.6%) out of 383 participants usually practiced outdoor PA. These data are in contrast with previous studies that found that parks were generally underutilized to perform PA [19,26]. In this case, the park seemed to be a facilitator of PA, due to the large numbers of people that used it as a training environment. However, even in this study, the PA performed in the park was not sufficient to achieve the goal of 150 min/week. In fact, when only PA in the park was considered, the majority of the sample did not achieve the goal of 150 min/week. This could mean that people did not do enough PA in the park, as reported in previous studies, showing that parks are more of a destination for light activities and low levels of PA, rather than a venue for moderate or vigorous PA [19,26], perhaps due to insufficient education regarding PA [19]. So, even if performing PA is one of the main motivations that drives people to use the park, the level or the intensity of PA were still not sufficient. Moreover, considering the third purpose of this study (to assess how active commuting influences the achievement of recommended levels of PA), the scenario changes. In fact, considering active commuting as a part of the PA, an increasing number of people who achieved the goal of recommended PA was observed. Active commuting is an important aspect to consider for the daily level of PA. Incorporating PA into daily life habits may make it easier to be physically active [64]. To our knowledge, even though the importance of active commuting is well established, few studies have analyzed together the minutes to reach the place of training by active commuting and the minutes of PA practiced. This result is particularly interesting because, while the PA carried out in green urban space alone is not enough to reach the goal of 150 min/week, the combination of this PA with active commuting makes it possible to reach this goal.

From the multiple regression, it emerged that the number of visits per week t at the park had a positive relationship with the amount of PA. If people have to walk more times in a week to reach the park, obviously their amount of PA increases. Regarding the way to reach the park, running was mostly associated with the total minutes of PA, while the use of motor vehicles presented a negative association. In any case, it is interesting to note that only 13.6% (n = 52) of the participants reached the park by motor vehicle. This pattern depends on the good walkability to reach the Arcoveggio park, thus providing evidence in favor of Bologna city policies.

This study is not without limits: the anthropometric measurements were self-reported by participants, and, in addition, this study was carried out only in one park in the city of Bologna, so the continuation of the study in other city parks may lead to a better understanding of the analyzed aspects. In addition, behavioral heterogeneity was not considered in the present study. For future research, it could be an important issue to consider [66,67,68].

This study also has numerous strengths. There is an increasing interest in active commuting and its importance in combination with PA. Active commuting is not only considered “an active way” to reach a place but it is considered a part of PA. To our knowledge, there have been no similar studies in the Italian setting, and this could be an important forerunner for future research.

## 5. Conclusions

Understanding the motivations that lead people to use an urban park is fundamental, since this allows the design of an appropriate project to encourage the use of this environment. At the same time, it is important to create a successful strategy to help the population to achieve the recommended levels of PA. The results of the present study suggest that there are two main reasons to visit the park, to relax and to practice PA. These are important aspects because they are useful to create appropriate projects or events to improve the use of green urban space. People showed that they were influenced by the restorative effects of the park and understood the importance of performing PA. Regarding PA, the people in the present study did not reach the minimum levels recommended by the WHO inside the park, but when active commuting was added, more people achieved the goal. This indicates the great importance of active transportation in an urban environment. Urban dwellers are largely physically inactive, and active methods of transport could be an important helper to include in their daily life habits. This represents an important point of reflection and may suggest the need to promote active commuting to raise awareness of the population about this important topic. In addition, it could be helpful to create a type of program for urban park users, such as putting signs inside the park, with tips and information about different kinds of outdoor exercise or examples of possible exercises (with explanatory drawings), facilitating the practice of more intensive PA.

## Figures and Tables

**Figure 1 ijerph-19-09248-f001:**
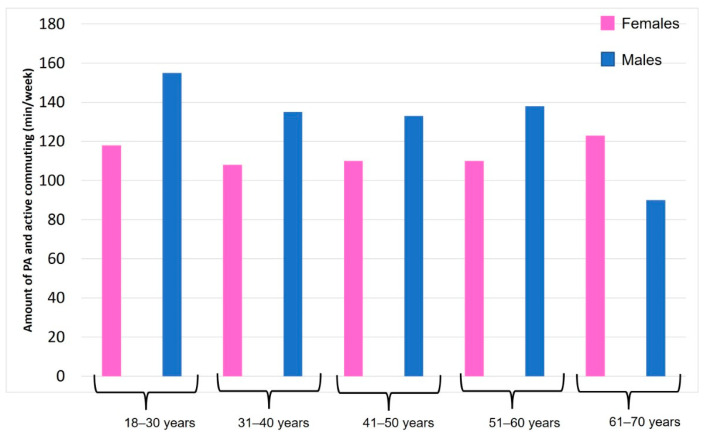
Amount of PA and active commuting in the park by sex and age classes.

**Table 1 ijerph-19-09248-t001:** Correlation values and p-value calculated for the validation of the questionnair.

Questions	Correlation Value	*p*-Value
Q1. Way to reach the park	0.74	0.00
Q2. Active commuting (min/week)	0.98	0.00
Q3. From 1 to 10, how tired were you when you reached the park?	0.68	0.00
Q4. How far (in meters, approximately) is your home from the park?	0.99	0.00
Q5. Do you go to the park to practice PA?	0.83	0.00
Q6. If you do not practice PA at the park, why do you go to the park?	0.70	0.00
Q7. Kind of PA	0.86	0.00
Q8. How many times per week do you go to the park?	0.88	0.00
Q9. How many hours of PA do you practice at the park?	0.93	0.00
Q10. If there wasn’t this park, would you have practiced PA in an indoor environment? How often would you practice indoor PA if there wasn’t a park (h/week)?	0.85 0.71	0.00 0.04
Q11. Where do you like to exercise the most?	0.88	0.00
Q12. If you practice indoor exercise, in which type of indoor environment?	0.80	0.00
Q13. Are you satisfied with this park?	0.76	0.00
Q14. I feel more energetic, after practicing PA in the park	0.75	0.00
Q15. I feel more energetic, after visiting the park	0.81	0.00

**Table 2 ijerph-19-09248-t002:** Anthropometric characteristics of participants (*n* = 383) by sex and age classes.

	Females (*n* = 215)					Males (*n* = 168)				
**Age (years)**	**18–30**	**31–40**	**41–50**	**51–60**	**61–70**	**18–30**	**31–40**	**41–50**	**51–60**	**61–70**
**N**	70	36	43	34	32	60	39	24	31	14
**%**	32.6	16.7	20	15.8	14.9	35.7	23.2	14.3	18.5	8.3
**Weight (kg)**	58.2	68.1	66.9	78.3	74.7	63.4	67.8	73.3	76.1	85.9
**SD**	9.2	16.6	13.5	12.6	10.6	10.3	13.4	11.7	9.4	12.5
**Height (cm)**	165.8	164.8	164.1	164.2	162.4	177.9	178.5	175.8	175	177.4
**SD**	8.1	6.7	5.3	7.2	5.2	8	5.1	7.1	7.7	7.9
**BMI (kg/m^2^)**	21.2	25.4	25.3	25	25.3	23.1	24.5	24.6	24.4	27.2
**SD**	2.8	3.8	5.8	4	4.8	2.9	3.1	2.7	2.8	3.5
**Weight status (%)**										
Underweight	17.3	11.2	4.7	-	3.1	5	-	-	-	-
Normal weight	75.6	69.5	53.5	56	59.4	73.3	64.1	66.6	70.9	21.4
Overweight	4.3	5.5	27.8	25.6	28.1	20	30.7	33.4	22.6	55.2
Obese	2.8	13.8	14	18.4	9.4	1.7	5.2	-	6.5	21.4

N (%) for categorical data; mean and standard deviation (SD) for continuous data.

**Table 3 ijerph-19-09248-t003:** Sociodemographic characteristics and questionnaire responses of participants (n = 383): comparisons among age groups and sexes.

	Females						Males						Females	Males	
	18–30 years	31–40 years	41–50 years	51–60 years	61–70 years	*p*-Value	18–30 years	31–40 years	41–50 years	51–60 years	61–70 years	*p*-Value	18–70 years	18–70 years	*p*-Value
**N (%)**	70 (32.6)	36 (16.7)	43 (20.0)	34 (15.8)	32 (14.9)		60 (35.7)	39 (23.2)	24 (14.3)	31 (18.5)	14 (8.3)		215 (56.1)	168 (43.9)	
**Occupation**						**0.00**						**0.00**			**0.00**
Employed	61.5	91.6	95.4	79.5	18.7	0.00	75.0	94.9	95.9	96.8	28.6	0.12	67.9	82.7	0.00
Student	37.1	2.8	-	-	-	0.00	16.7	-	-	-	-	0.00	16.3	6.0	0.00
Unemployed	1.4	5.6	4.6	11.7	-	0.29	8.3	5.1	-	3.2	-	0.00	3.7	4.8	0.00
Retired	-	-	-	8.8	81.3	0.00	-	-	4.1	-	71.4	0.00	12.1	6.5	0.00
**Education level**						**0.06**						**0.07**			**0.23**
High school	40.0	50.0	46.5	67.7	62.5	0.00	45.0	61.5	41.7	58.1	92.9	0.00	49.7	55.4	0.75
Bachelor’s degree	48.6	16.7	11.6	14.7	25.0	0.00	26.7	15.4	12.5	16.1	-	0.07	27.9	17.3	0.10
Master’s degree	10.0	30.6	32.6	17.6	9.4	0.01	28.3	15.4	41.7	22.6	7.1	0.01	19.1	24.4	0.53
PhD	1.4	2.7	9.3	-	3.1	0.00	-	7.7	4.1	3.2	-	0.00	3.3	2.9	0.98
**Q1. Way to reach the park**						**0.00**						**0.00**			**0.00**
Walking	52.8	83.3	60.4	79.4	65.6	0.02	40.0	43.5	45.8	54.8	50.0	0.00	65.6	56.7	0.01
Running	11.4	-	2.3	2.9	-	0.09	35.0	17.9	29.1	9.6	7.1	0.16	4.7	24.6	0.24
Bicycle	18.5	11.1	13.9	11.7	3.1	0.05	15.0	33.3	20.8	25.8	14.2	0.00	13.0	11.2	0.00
Vehicle	17.1	5.6	23.3	5.8	31.2	0.74	10.0	5.1	4.1	9.6	28.5	0.04	16.7	7.5	0.04
**Q5. Do you go to the park to practice PA?**						**0.29**						**0.01**			**0.00**
No	41.4	50	55.8	55.9	62.5	0.04	20.0	23.1	16.7	25.8	64.3	0.04	51.2	25.0	0.00
Yes	58.6	50	44.2	44.1	37.5	0.15	80.0	76.9	83.3	74.2	35.7	0.00	48.8	75.0	0.00
**Q6. If you do not practice PA at the park, why do you go to the park?**						**0.01**						**0.15**			**0.60**
Get in touch with nature	11.9	19.4	29.3	25.8	16.7	0.02	7.4	14.3	13.0	7.7	53.8	0.00	19.8	17.3	0.60
Relax	67.8	74.2	46.3	61.3	83.3	0.20	77.8	67.9	73.9	92.3	30.8	0.48	65.6	70.7	0.57
Socializing	20.3	6.5	24.4	12.9	-	0.00	14.8	17.9	13.0	-	15.4	0.16	14.6	12.0	0.52
**Q7. Kind of PA**						**0.00**						**0.00**			**0.00**
Light running	23.2	18.7	10.6	-	-	0.40	18.2	24.2	36.8	22.7	14.3	0.00	7.0	17.9	0.07
Outdoor fitness equipment	7.7	-	10.6	-	7.1	0.01	68.2	27.6	-	-	-	0.00	5.5	36.9	0.00
Skating	3.7	-	10.6	-	-	0.00	-	-	-	-	-	0.05	1.3	-	0.27
Walking	38.7	81.3	47.2	100	71.4	0.03	4.5	24.2	42.1	59.1	57.1	0.00	75.3	32.3	0.00
Football	-	-	-	-	7.1	0.00	2.3	10.4	-	-	-	0.00	1.3	1.2	0.90
Bicycling	19.3	-	5.2	-	-	0.02	-	13.7	-	18.2	28.6	0.00	7.0	8.3	0.54
Stretching	3.7	-	10.6	-	-	0.00	6.9	-	10.5	-	-	0.00	1.3	2.4	0.67
Nordic walking	3.7	-	5.2	-	14.4	0.42	-	-	10.5	-	-	0.00	1.3	1.0	0.28
**Q10. If there wasn’t this park, would you have practiced PA in an indoor environment?**						**0.01**						**0.00**			**0.75**
Yes	51.4	50.0	37.2	32.4	21.9	0.19	58.3	28.2	66.7	29.0	21.4	0.02	40.9	44.0	0.89
No	38.6	25.0	39.5	55.9	68.8	0.05	31.7	59.0	33.3	51.6	42.9	0.27	43.7	42.9	0.99
I don’t know	10.0	25.0	23.3	11.8	9.4	0.14	10.0	12.8	-	19.4	35.7	0.05	15.4	13.1	0.84
**Q11. Where do you like to exercise the most?**						**0.01**						**0.03**			**0.00**
Outdoor	77.1	82.9	95.3	94.1	90.6	0.79	91.5	89.7	91.7	93.5	91.7	0.08	86.9	91.5	0.63
Indoor	22.9	17.1	2.3	5.9	9.4	0.00	8.5	10.3	8.3	6.5	8.3	0.09	13.1	8.5	0.00
**Q12. If you practice indoor exercise, in which type of indoor environment?**						**0.53**						**0.19**			**0.75**
Home	37.9	28.1	28.9	46.4	47.8	0.56	35.8	37.5	29.2	32.0	27.3	0.56	39.9	33.8	0.64
Gym	59.1	59.4	63.2	50.0	52.2	0.95	62.3	56.3	70.8	48.0	54.5	0.86	57.8	59.3	0.85
Swimming pool	3.0	12.5	7.9	3.6	-	0.28	1.9	6.3	-	20.0	18.2	0.01	2.3	6.9	0.57

Note. Differences between the overall categories are reported in bold.

**Table 4 ijerph-19-09248-t004:** Descriptive statistics and ANOVA by sex, age groups, and interaction between sexes and age groups.

	Females					Males					ANOVA					
	18–30 yrs	31–40 yrs	41–50 yrs	51–60 yrs	61–70 yrs	18–30 yrs	31–40 yrs	41–50 yrs	51–60 yrs	61–70 yrs	Age Class		Sex		Age Class * Sex	
Variable	Mean (SD)	Mean (SD)	Mean (SD)	Mean (SD)	Mean (SD)	Mean (SD)	Mean (SD)	Mean (SD)	Mean (SD)	Mean (SD)	F	*p*	F	*p*	F	*p*
**Q2. Active commuting (min/week)**																
	72.85 (48.25)	65.13 (42.27)	68.61 (50.63)	56.41 (46.67)	64.18 (43.41)	81.80 (47.31)	76.12 (56.43)	53.41 (42.26)	68.14 (34.00)	75.86 (38.80)	1.58	0.17	0.77	0.37	0.82	0.50
**Q3. From 1 to 10, how tired were you when you reached the park?**																
	2.33 (0.88)	1.94 (0.75)	1.57 (0.81)	2.53 (1.43)	2.58 (1.08)	1.53 (0.67)	1.42 (0.51)	1.13 (0.35)	1.10 (0.32)	1.40 (0.55)	2.45	**0.04**	34.64	**0.00**	1.57	0.18
**Q4. How far (in meters, approximately) is your home from the park?**																
	2971.4 (3034.0)	2582.7 (5512.7)	2110.2 (4662.4)	1962.1 (2344.3)	2422.8 (2024.0)	2631.6 (2819.9)	2311.5 (1265.6)	1706.2 (1903.7)	3220.9 (3910.2)	2342.3 (3001.3)	0.80	0.53	0.00	0.97	0.76	0.55
**Q8. How many times per week do you go to the park?**																
	2.51 (1.54)	3.22 (1.57)	3.37 (1.62)	3.53 (1.95)	2.33 (1.22)	2.71 (1.44)	2.35 (1.42)	3.06 (1.59)	2.60 (1.57)	3.75 (1.94)	2.07	0.08	0.24	0.61	4.96	**0.00**
**Q9. How many hours of PA do you practice at the park?**																
	1.33 (0.61)	1.33 (0.49)	1.29 (0.73)	1.47 (0.64)	1.83 (1.70)	1.65 (0.63)	1.27 (0.45)	1.63 (0.67)	1.35 (0.57)	1.20 (0.45)	1.16	0.32	3.48	0.06	1.39	0.23
**Q10. How often would you practice indoor PA if there wasn’t a park (h/week)?**																
	2.67 (1.12)	2.11 (0.76)	1.94 (0.44)	1.91 (0.30)	2.00 (0.00)	2.57 (1.07)	2.45 (0.82)	2.19 (0.66)	2.11 (0.60)	2.67 (0.58)	3.42	**0.01**	2.50	0.11	0.65	0.62
**Q11. Where do you like to exercise the most?**																
Indoor	2.54 (0.94)	3.19 (0.98)	2.90 (0.55)	2.55 (0.82)	2.14 (0.90)	2.30 (0.88)	2.48 (0.73)	2.63 (1.06)	2.00 (0.85)	3.00 (0.00)	2.40	**0.05**	0.97	0.32	1.34	0.25
Outdoor	3.00 (1.38)	2.89 (1.41)	2.04 (1.46)	2.05 (1.47)	2.50 (1.54)	2.58 (1.50)	2.66 (1.52)	2.50 (1.55)	2.56 (1.53)	2.71 (1.60)	1.51	0.19	0.08	0.77	1.12	0.34
**Q13. Are you satisfied with this park?**																
	4.20 (0.83)	4.11 (0.87)	4.42 (0.88)	4.29 (1.14)	4.38 (0.75)	3.78 (0.99)	4.08 (1.16)	4.42 (0.58)	4.50 (0.63)	3.77 (1.59)	3.33	**0.01**	2.12	0.14	1.75	0.13
**Q14. I feel more energetic, after practicing PA in the park**																
	3.21 (1.36)	3.72 (1.32)	3.57 (1.35)	3.94 (1.58)	3.34 (1.73)	3.08 (1.34)	3.10 (1.65)	3.25 (1.36)	3.45 (1.71)	3.46 (1.66)	0.51	0.19	2.52	0.11	0.69	0.59
**Q15. I feel more peaceful, after visiting the park**																
	3.35 (1.33)	3.69 (1.37)	3.55 (1.43)	4.09 (1.36)	3.69 (1.53)	3.98 (1.25)	3.35 (1.34)	3.88 (1.45)	3.32 (1.28)	3.85 (1.46)	0.24	0.91	0.02	0.87	3.32	**0.01**
**Amount of PA (min/week)**																
	79.76 (36.50)	80.00 (29.10)	77.37 (43.95)	88.00 (38.40)	110.00 (101.80)	98.75 (37.62)	76.00 (26.99)	97.50 (39.98)	80.87 (34.37)	72.00 (26.83)	0.62	0.65	0.09	0.76	2.18	0.07
**PA and active commuting (min/week)**																
	117.12 (73.32)	107.49 (63.33)	111.08 (76.15)	111.46 (78.14)	125.00 (91.51)	155.20 (69.21)	134.12 (75.18)	132.44 (63.95)	139.30 (62.09)	92.90 (55.30)	1.16	0.32	3.48	0.06	1.39	0.23

Note. F = F test, *p* = *p*-value, * = interaction between the two variables, yrs = years.

**Table 5 ijerph-19-09248-t005:** Multiple regression model for total minutes of PA.

Predictors	β	T	*p*-Value
Sex (female)	–0.11	−0.12	0.90
**BMI**	0.04	0.51	0.61
**Distance from the park**	0.15	1.56	0.12
**Times per week at the park**	0.18	2.12	**0.04**
**Age class**			
18–30 years	–0.01	–0.08	0.93
31–40 years	0.01	0.06	0.95
41–50 years	–0.01	–0.11	0.92
51–60 years	0.04	0.38	0.17
**Occupation**			
Student	–0.05	–0.45	0.66
Employed	0.07	0.55	0.58
Unemployed	–0.17	–1.45	0.15
**Education level**			
High school	0.01	0.02	0.98
Bachelor’s degree	0.07	0.87	0.38
Master’s degree	0.08	0.93	0.36
**Way to reach the park**			
Walking	0.14	1.19	0.24
Motor vehicle	–0.36	–2.94	**0.04**
Running	0.22	1.98	**0.04**
**Kind of PA at the park**			
Light running	–0.13	–1.36	0.18
Outdoor fitness equipment	0.07	0.64	0.52
Skating	0.30	2.95	**0.03**
Walking	0.28	2.17	**0.03**
Football	0.06	0.68	0.49
Bicycling	−0.07	−0.67	0.50
Stretching	0.04	0.47	0.64
*R^2^*	*0.28*		
*Adjusted R^2^*	*0.14*		
*p*	** *0.01* **		

## Data Availability

Authors will provide data to all interested parties upon reasonable request.

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
