# Peer review of "Physical Activity Behavior, Motivation and Active Commuting: Relationships with the Use of Green Spaces in Italy"

_ijerph, 2022, doi:10.3390/ijerph19159248_

Round 1
Reviewer 1 Report
The Authors decided to analyze the subject of physical activity as the part of the lifestyle in the big city. The problem is very important and it is very good that the Authors developed this study. The manuscript is well written. The study has its limitations, which are pointed out in the limitations of the study paragraph. The study is the observational type, so it also impacts its reliability. None the less, I believe that this work could contribute to future exploration of this subject.
I would like to make one main comment regarding the results described in the paper.
In the table 2, the Authors presented p value only for the whole categories (e.g.”occupation”) instead of subcategories. In such presentation, what does the p value refer to – differences between subcategories (eg. “retired” vs. “employed”) or between the age groups within the same category? Please explain this and fill in the p values missing in the other lines of the table.
Author Response
The Authors decided to analyze the subject of physical activity as the part of the lifestyle in the big city. The problem is very important and it is very good that the Authors developed this study. The manuscript is well written. The study has its limitations, which are pointed out in the limitations of the study paragraph. The study is the observational type, so it also impacts its reliability. None the less, I believe that this work could contribute to future exploration of this subject.
I would like to make one main comment regarding the results described in the paper.
Answer. Firstly, we greatly appreciate all the time dedicated, as well as your significant comments. All your suggestions have been considered in the new version of the manuscript. We strongly believe that our manuscript has improved, largely due to the highlighted considerations. Thank you so much for this. Check out the manuscript changes answered point by point below.
In the table 2, the Authors presented the p value only for the whole categories (e.g.”occupation”) instead of subcategories. In such presentation, what does the p value refer to – differences between subcategories (eg. “retired” vs. “employed”) or between the age groups within the same category? Please explain this and fill in the p values missing in the other lines of the table.
A. Thank you for the suggestion, the p-value for each subcategory was added to the table. Moreover, the p-value in the first row of each category refers to the difference in the distribution in the subcategories among the age classes.
Reviewer 2 Report
The work addresses a new and current topic, especially after the COVID-19 pandemic.
In general, the study is well done. Some recommendations are indicated for its improvement (the 2 and 7 are very important).
1.- There is a certain lack of relationship between objective 3 and objectives 1 and 2.
It is recommended to provide more coherence by revising the Introduction section.
2.- It is necessary to provide more information on the validation process of the questionnaire and its results.
3.- Figure 1. What do the numbers 1-10 mean? Are these numbers really necessary?
4.- The format of the tables should be improved. For example:
In Table 2, the first rows are written over other words and it doesn't read well.
In Table 3, the information does not appear centered in each column. That makes it hard to read
5.- Line 212: It indicates: “Generally, men practiced more PA than women”.
It is recommended to specify more, that is, indicate the cases in which it is not so.
6.- Line 263: Why this data is in contrast with the study by de Vries et al. (2003)? What is the reason for these different results?
7.- Does the data violate multiple linear regression assumptions? Nothing is said about it. it is necessary to indicate how they have been verified and the results. This is the major limitation of this study.
Author Response
The work addresses a new and current topic, especially after the COVID-19 pandemic.
In general, the study is well done. Some recommendations are indicated for its improvement (the 2 and 7 are very important).
Answer. Firstly, we greatly appreciate all the time dedicated, as well as your significant comments. All your suggestions have been considered in the new version of the manuscript. We strongly believe that our manuscript has improved, largely due to the highlighted considerations. Thank you so much for this. Check out the manuscript changes answered point by point below.
1.- There is a certain lack of relationship between objective 3 and objectives 1 and 2.
It is recommended to provide more coherence by revising the Introduction section.
A. Thank you for your suggestion. We have clarified the relationship between objective 3 and objectives 1 and 2 (lines 58, 73-77, 85-86).
2.- It is necessary to provide more information on the validation process of the questionnaire and its results.
A. Thank you for the comment. More information on the validation process of the questionnaire was added in lines 129-130, 156-160, and Table 1 was added.
3.- Figure 1. What do the numbers 1-10 mean? Are these numbers really necessary?
A. Thank you for the comment, the numbers 1-10 were a typo of an older version, so we removed them.
4.- The format of the tables should be improved. For example:
In Table 2, the first rows are written over other words and it doesn't read well.
In Table 3, the information does not appear centered in each column. That makes it hard to read
A. Thank you for your suggestion, the format of the tables has been improved.
5.- Line 212: It indicates: “Generally, men practiced more PA than women”.
It is recommended to specify more, that is, indicate the cases in which it is not so.
A. Thank you for the comment, we indicated the case in which women practiced more PA than men (lines 240-243).
6.- Line 263: Why this data is in contrast with the study by de Vries et al. (2003)? What is the reason for these different results?
A. Thank you for your concern. A sentence was added in the discussion part (lines 318-320)
7.- Does the data violate multiple linear regression assumptions? Nothing is said about it. it is necessary to indicate how they have been verified and the results. This is the major limitation of this study.
A. Thank you for the suggestion, we added some information about the multiple regression in lines 146-148, and lines 282-283.
Reviewer 3 Report
Review report for IJERPH paper
This paper designed a questionnaire and developed a multiple regression method for understanding the motivations which lead people to use an urban park for physical activity. The results in this study shows that there are two main reasons to reach the park: to relax and to practice PA. Some comments are as follows to improve the paper:
1. The contributions of this study are not clear. The questionnaire is well-designed, however, how to assess people’s motivation to use park, and how to value the association between park use and levels of PA are still not clear in the conclusion of this paper.
2. “Furthermore, the recent emergence of SARS-CoV-2 has influenced the lifestyle of the population, reducing PA, and becoming a serious concern mainly for older adults who are typically more prone to chronic diseases and less active compared to younger people [5] (Anwari et al. 2021; Yao et al. 2022).” The statement needs evidence and supports. The two references are appropriate for supporting the statement.
Anwari, Nafis, Md Tawkir Ahmed, Md Rakibul Islam, Md Hadiuzzaman, and Shohel Amin. "Exploring the travel behavior changes caused by the COVID-19 crisis: A case study for a developing country." Transportation Research Interdisciplinary Perspectives 9 (2021): 100334.
Yao, W., Yu, J., Yang, Y., Chen, N., Jin, S., Hu, Y., and Bai, C., 2022. Understanding travel behavior adjustment under COVID-19. Communications in Transportation Research, 2, 100068. doi:10.1016/j.commtr.2022.100068.
3. In Table 1, the authors summarized the anthropometric characteristics of the study participants. What is the conclusion of Table 1, and what is the relationship between physical activity and the variables in Table 1 like height, BMI or weight status?
4. Some question settings are inappropriate. For example, in Table 2 Q1, the authors want to know the relationship between “the way to reach the park” and “the sex and age classes”, however, the travel mode is relevant to the accessibility and the distance between home and park.
5. In page 5, line 3, the author claims that “Most of the participants preferred outdoor PA (88.9%), probably because outdoor PA was considered safer than gym training during the Covid-19 pandemic”. Is there any before-after study or any data support this statement?
6. There are large variations in the behaviors among different people (Xu et al. 2021; Ortúzar 2021). How do you consider this issue? Corresponding clarifications should be added. If the behaviroal heterogeneity is not considered, at least, a corresponding limitation statement should be provided to clarify this by adding the references.
A. Stathopoulos, S. Hess, Revisiting reference point formation, gains–losses asymmetry and nonlinear sensitivities with an emphasis on attribute specific treatment
Ortúzar, J. de D. (2021) ‘Future transportation: Sustainability, complexity and individualization of choices’
7. In page 11, Table 4 shows that the statistically significant variables are “times per week at the park”, “way to reach the park” and “kind of PA at he park”. Are these variables sufficient to reveal the association between park use and levels of PA?
8. It seems that the authors are concerned about the COVID-19 pandemic. How is it reflected in the question settings and option settings?
9. Minor concerns:
1) Page number error from Page 10 to Page 17
2) Table 2 and Table 3 are too large and can be simplified.
3) It is better to distinguish the RP and SP survey.
Author Response
This paper designed a questionnaire and developed a multiple regression method for understanding the motivations which lead people to use an urban park for physical activity. The results in this study shows that there are two main reasons to reach the park: to relax and to practice PA. Some comments are as follows to improve the paper:
Answer (A). Firstly, we greatly appreciate all the time dedicated, as well as your significant comments. All your suggestions have been considered in the new version of the manuscript. We strongly believe that our manuscript has improved, largely due to the highlighted considerations. Thank you so much for this. Check out the manuscript changes answered point by point below.
1. The contributions of this study are not clear. The questionnaire is well-designed, however, how to assess people’s motivation to use park, and how to value the association between park use and levels of PA are still not clear in the conclusion of this paper.
A. Thank you for the comment, we clarified this point in the discussion (lines 354-358, 366-368).
2. “Furthermore, the recent emergence of SARS-CoV-2 has influenced the lifestyle of the population, reducing PA, and becoming a serious concern mainly for older adults who are typically more prone to chronic diseases and less active compared to younger people [5] (Anwari et al. 2021; Yao et al. 2022).” The statement needs evidence and supports. The two references are appropriate for supporting the statement.
Anwari, Nafis, Md Tawkir Ahmed, Md Rakibul Islam, Md Hadiuzzaman, and Shohel Amin. "Exploring the travel behavior changes caused by the COVID-19 crisis: A case study for a developing country." Transportation Research Interdisciplinary Perspectives 9 (2021): 100334.
Yao, W., Yu, J., Yang, Y., Chen, N., Jin, S., Hu, Y., and Bai, C., 2022. Understanding travel behavior adjustment under COVID-19. Communications in Transportation Research, 2, 100068. doi:10.1016/j.commtr.2022.100068.
A. Thank you for your suggestion. We have added the suggested references.
3. In Table 1, the authors summarized the anthropometric characteristics of the study participants. What is the conclusion of Table 1, and what is the relationship between physical activity and the variables in Table 1 like height, BMI or weight status?
A. Thank you for your concern, an explanation was added in the discussion part (lines 326-331).
4. Some question settings are inappropriate. For example, in Table 2 Q1, the authors want to know the relationship between “the way to reach the park” and “the sex and age classes”, however, the travel mode is relevant to the accessibility and the distance between home and park.
A. Thank you for the comment. The chi-squared test for the distance between home and the park and the way to reach the park was added (lines 228-233). We decided to also maintain the previous version and to consider the relationship between the way to reach the park and sex and age classes since it can be an added value compared to other studies.
5. In page 5, line 3, the author claims that “Most of the participants preferred outdoor PA (88.9%), probably because outdoor PA was considered safer than gym training during the Covid-19 pandemic”. Is there any before-after study or any data support this statement?
A. Thank you for your concern. We have added the appropriate references.
6. There are large variations in the behaviors among different people (Xu et al. 2021; Ortúzar 2021). How do you consider this issue? Corresponding clarifications should be added. If the behaviroal heterogeneity is not considered, at least, a corresponding limitation statement should be provided to clarify this by adding the references.
Stathopoulos, S. Hess, Revisiting reference point formation, gains–losses asymmetry and nonlinear sensitivities with an emphasis on attribute specific treatment
Ortúzar, J. de D. (2021) ‘Future transportation: Sustainability, complexity and individualization of choices’
A. Thank you for the comment and suggestions, we added this issue to the limitation of the study (lines 391-393).
7. In page 11, Table 4 shows that the statistically significant variables are “times per week at the park”, “way to reach the park” and “kind of PA at the park”. Are these variables sufficient to reveal the association between park use and levels of PA?
A. By the regression we individuated the more informative variables for this study, but future studies will help to better understand the association between park use and levels of PA.
8. It seems that the authors are concerned about the COVID-19 pandemic. How is it reflected in the question settings and option settings?
A. Since the questionnaire was administered during the COVID-19 pandemic (March 2021-April 2021), we think that the answers could be influenced by the specific historical period. Unfortunately, specific questions regarding the COVID-19 pandemic were not inserted in the questionnaire since it was not the specific topic of the survey.
- Minor concerns:
1) Page number error from Page 10 to Page 17
2) Table 2 and Table 3 are too large and can be simplified.
3) It is better to distinguish the RP and SP survey.
A. Thank you for the suggestions, we will ask the Editor how to fix the problem with the number of pages. We know that the two tables are very large, but we think that all the data reported are important to better understand the results of the present study. A sentence was added about the RP and SP survey (lines 110-113).
Round 2
Reviewer 3 Report
Thank the authors for addressing my comments.